# Simultaneous Electrochemical Detection of Dopamine and Tryptophan Using 3D Goethite–Spongin Composites

**DOI:** 10.3390/biomimetics9060357

**Published:** 2024-06-14

**Authors:** Sedigheh Falahi, Anita Kubiak, Alona Voronkina, Hermann Ehrlich, Yvonne Joseph, Parvaneh Rahimi

**Affiliations:** 1Institute of Nanoscale and Biobased Materials, Faculty of Materials Science and Material Technology, Technische Universität Bergakademie Freiberg, 09599 Freiberg, Germany; sedigheh.falahi@doktorand.tu-freiberg.de (S.F.); yvonne.joseph@esm.tu-freiberg.de (Y.J.); 2Faculty of Chemistry, Adam Mickiewicz University, Uniwersytetu Poznanskiego 8, 61-614 Poznan, Poland; anikub@amu.edu.pl (A.K.); isididae@gmail.com (H.E.); 3Center of Advanced Technology, Adam Mickiewicz University, Uniwersytetu Poznanskiego 10, 61-614 Poznan, Poland; 4Department of Pharmacy, National Pirogov Memorial Medical University, Vinnytsya, Pyrogov Street 56, 21018 Vinnytsia, Ukraine; voronkina@vnmu.edu.ua; 5Faculty of Chemical Technology, Institute of Chemical Technology and Engineering, Poznan University of Technology, Berdychowo 4, 60-965 Poznan, Poland; 6Freiberg Water Research Center, Technische Universität Bergakademie Freiberg, 09599 Freiberg, Germany

**Keywords:** electrochemical, simultaneous determination, bio-nanocomposite, dopamine, tryptophan, spongin, goethite

## Abstract

In this study, a facile approach for simultaneous determination of dopamine (DA) and tryptophan (TRP) using a 3D goethite–spongin-modified carbon paste electrode is reported. The prepared electrode exhibited excellent electrochemical catalytic activity towards DA and TRP oxidation. The electrochemical sensing of the modified electrode was investigated using cyclic voltammetry, differential pulse voltammetry, and electrochemical impedance spectroscopy. Through differential pulse voltammetry analysis, two well-separated oxidation peaks were observed at 28 and 77 mV, corresponding to the oxidation of DA and TRP at the working electrode, with a large peak separation of up to 490 mV. DA and TRP were determined both individually and simultaneously in their dualistic mixture. As a result, the anodic peak currents and the concentrations of DA and TRP were found to exhibit linearity within the ranges of 4–246 μM for DA and 2 to 150 μM for TRP. The detection limits (S/N = 3) as low as 1.9 μM and 0.37 μM were achieved for DA and TRP, respectively. The proposed sensor was successfully applied to the simultaneous determination of DA and TRP in human urine samples with satisfactory recoveries (101% to 116%).

## 1. Introduction

Quantitative assessment of amino acids is essential in numerous fields, such as bio-engineering, bio-medical investigations, and food monitoring [1]. Dopamine (DA) is a primary monoamine neurotransmitter in the brain which is significantly produced by mesenteric organs [2]. It plays a vital role in higher brain functions, including motivation [3], arousal [4], spatial memory [5], and regulating motor neurons [6]. Given the extensive and crucial roles of DA, abnormal DA metabolism is associated with the development of various mental disorders, including Huntington’s disease (HD), schizophrenia, Parkinson’s disease (PD), addiction, and attention-deficit/hyperactivity disorder (ADHD) [7]. Tryptophan (TRP), a vital amino acid, is an essential component of the human diet. It plays a key role in protein synthesis and is the precursor for serotonin, melatonin (ML), nicotinamide (vitamin B6), kynurenine, and niacin [8,9]. An increased metabolism of TRP, or adverse effects of low TRP, has been observed in different types of disease and disorders such as Parkinson’s disease [10], chronic kidney disease [11], mental disorders [12], sleep disorders [13], and major depressive disorder (MDD) [14]. Given that the human body cannot synthesize TRP independently, its incorporation into food and pharmaceutical products is necessary. Consequently, the development of methods that enable efficient, rapid, sensitive, specific, and cost-effective simultaneous detection of these biomolecules, which largely affect the behavioral and cognitive functions of human, is of considerable interest in diagnostics, especially for neurological disorders. Generally, analytical techniques such as high-performance liquid chromatography (HPLC) [15], fluorescence spectrometry [16], ultraviolet–visible (UV–vis) spectrophotometry [17] and capillary electrophoresis [18] are utilized for the detection of biological analytes. However, despite the high sensitivity of these techniques, they are often hindered by limitations, including high cost, complicated operation, and sample pretreatment requirements, making them undesirable for routine analysis applications. On the other hand, since DA and TRP are electroactive compounds, electrochemical approaches including cyclic voltammetry (CV), differential pulse voltammetry (DPV), and amperometry are the most reliable techniques for DA and TRP determination, offering high sensitivity, specificity, selectivity, low cost, rapid analysis, and potential for miniaturization. Negut et al. have comprehensively reviewed the recent trends in electrochemical detection of DA and TRP [19]. Direct electrochemical detection of these molecules at a bare electrode is very slow, which hinders the sensitivity and selectivity of the sensor and also necessitates a high potential for accurate determination [20]. Another challenge in the electrochemical detection of these molecules is the oxidation potential overlap of ascorbic acid (AA), and DA as well as electrode fouling caused by TRP. Therefore, when it comes to their simultaneous detection as multitaskers, electrode modifiers and working pH serve an important role in enhancing the sensor performance [21,22]. Carbon-based electrodes, including carbon paste electrodes (CPEs) [23,24,25], and glassy carbon electrodes (GCEs) [26,27,28], are commonly employed as working electrodes in the simultaneous determination of DA and TRP. In comparison with GCEs, CPEs are often preferred owing to their low cost, simple and fast fabrication method, quick renewable surface, and minimal background current [29]. Nevertheless, due to the lack of selectivity and sensitivity of CPEs towards the target analyte, it is possible to enhance their electrochemical performance by incorporating organic and inorganic compounds into the carbon paste [30]. To date, the majority of studies focusing on the simultaneous detection of DA and TRP have predominantly utilized carbon nanostructured materials. This preference is attributed to their high surface area, biocompatibility, and satisfactory conductivity, which are often further optimized through the incorporation of metal nanoparticles, metal oxides, or polymers. For instance, composites such as titanium dioxide/reduced graphene oxide (TiO_2_/RGO) [23], nickel(II) oxide/carbon nanotube/poly(3,4-ethylenedioxythiophene) (NiO/CNT/PEDOT) [26], gold/palladium/copper/RGO/multi-walled carbon nanotube (AuPdCu/RGO/MWCNT) [31], TiO_2_-graphene/poly(4-aminobenzenesulfonic acid) (TiO_2_-GR/4-ABSA) [32], and molecularly imprinted polymer [33] have been employed for detection of sub-micromolar concentrations of DA and TRP. Metal oxides offer many advantages in the electrochemical detection of DA and TRP; however, they also have certain disadvantages and limitations, including stability issues, complex fabrication process, and environmental impacts if using toxic chemicals during the synthesis process. Compared to alternative metal oxides, ferric oxyhydroxide (FeOOH) offers distinct advantages in terms of its affordability, ecological compatibility, and ease of synthesis. These attributes significantly expand its potential for application in a diverse range of electrochemical sensing applications [34,35,36,37,38]. In a study, Moolayadukkam et al. [39] investigated the phase dependence of three crystalline phases (α-, β-, and γ-) of FeOOH for electrochemical DA detection. Based on this study, utilizing GCE, a desirable sensitivity of 0.27, 0.33, and 0.20 µA·µM^−1^·cm^−2^ was reported for α-FeOOH (goethite), β-FeOOH (akaganeite), and γ-FeOOH (lepidocrocite), respectively. These results reveal the high catalytic performance of FeOOH towards DA oxidation. Additionally, incorporating ferric mineral-based nanostructures into various 2D and 3D carbon nanomaterials, polymers, and biopolymers can significantly enhance electrocatalytic kinetics and increase the specific surface areas [40,41,42,43]. Renewable biopolymers, such as spongin, which is the primary protein-based component of the cultivated bath sponges skeletons, are increasingly gaining attention in the field of advanced biomaterials science [44,45]. Up to date, spongin-based naturally occurring 3D scaffolds [46] have been successfully employed in extreme biomimetic [47], including for carbonization [48], for the development of new metal oxide [49,50,51], and mineral composite materials [52,53,54], electrocatalysts [55,56], and as an effective substrate for enzyme immobilization [57]. In our recent studies, using biomimetic approach, we have developed electrochemical sensors for DA detection based on lepidocrocite- [53] and goethite- [54] containing 3D spongin scaffolds. A high sensitivity of 0.14 and 0.21 µA·µM^−1^·cm^−2^, respectively, has been obtained for these sensors.

In the present study, a cost-effective, environmentally friendly, and sensitive electrochemical sensor for simultaneous determination of DA and TRP is developed. The sensor was constructed based on a goethite-containing 3D spongin scaffold (denoted as 3DGS) owing to its unique electronic and physical features. The prepared 3DGS composite was utilized for modification of CPE and was further advanced for simultaneous electrochemical detection of DA and TRP using CV and DPV methods. Furthermore, the fabricated electrochemical sensor was evaluated for its potential to detect DA and TRP in human urine samples.

## 2. Materials and Methods

### 2.1. Reagents

Graphite powder, DA, TRP, AA, ML, monosodium dihydrogen phosphate (NaH_2_PO_4_), disodium orthohydrogen phosphate (Na_2_HPO_4_), and paraffin oil were purchased from Sigma Aldrich (Burlington, MA, USA). Potassium ferricyanide and potassium ferrocyanide were obtained from Merck (Darmstadt, Germany). Iron powder (99.99%, with particle sizes in the range of 25–100 µM) and crystalline iodine (99.8%) were acquired from Chempur (Piekary Śląskie, Poland). Purified spongin scaffolds from the marine demosponge of *Hippospongia communis* (Lamarck, 1814) were purchased from INTIB GmbH (Freiberg, Germany). Uric acid (UA) was acquired from Alfa Aesar (Karlsruhe, Germany). Alanine and potassium chloride (KCl) were purchased from Carl Roth (Karlsruhe, Germany). Also, calcium chloride (CaCl_2_) and β-D(+) glucose were acquired from Fluka (Steinheim, Germany) and VWR Prolabo (Darmstadt, Germany), respectively. Sodium chloride (NaCl) was bought from Riedel-de-Haen, (Seelze, Germany). All solutions were prepared in double-distilled deionized water. Fresh human urine specimens were collected from the colleagues and utilized without any pretreatment for DA and TRP quantification.

### 2.2. Apparatus and Measurements

Electrochemical analysis was conducted utilizing a PalmSens 4 electrochemical analyzer, operated by PSTrace 5.8 software (PalmSens BV, Houten, The Netherlands). The experiments were performed under standard room temperature conditions in a three-electrode setup. A saturated silver/silver chloride (Ag/AgCl in 3 M KCl solution) and a platinum wire electrode were used as a reference electrode and an auxiliary electrode, respectively. Fabricated CPEs served as the working electrodes. Phosphate buffer solution (0.1 M) ranging from pH 3.0 to 8.0 was prepared using a mixture of the NaH_2_PO_4_ and Na_2_HPO_4_ stock solutions and employed as electrolyte buffer solution for all measurements. EIS was conducted in a solution of 0.1 M KCl containing 5 mM [Fe(CN)_6_]^3−/4−^. The frequency range examined was from 0.1 Hz to 100 kHz, with an applied potential of 0.22 V and an amplitude of 5 mV.

### 2.3. Preparation of the Electrodes

3DGS was prepared according to our earlier report [54]. The 3DGS samples were finely ground using a mortar and pestle under liquid nitrogen. The modified CPEs were fabricated by grinding the optimized ratios of 0.35 g graphite powder, 0.05 g 3DGS powder, and 120 µL paraffin oil as a binder in a mortar with a grinding time of 40 min. The components were homogenized to form a paste, which was then pressed into a carbon paste holder with an inner diameter of 4 mm. The unmodified CPE was prepared in the same manner without 3DGS powder. These electrodes are denoted as CPE (bare CPE) and 3DGS/CPE (3DGS-modified CPE). The surface of the electrodes was mechanically renewed by polishing it on the weighing paper before starting a new experiment.

## 3. Results and Discussion

### 3.1. Electrochemical Characterization of Prepared Electrodes

EIS was employed for investigating the electrochemical features of CPE and 3DGS/CPE in 0.1 M KCl containing 5 mM [Fe(CN)_6_]^3−/4−^ at the frequency range of 0.1 Hz–100 kHz, an applied potential of 0.22 V, and an amplitude of 5 mV (Figure 1A). The impedance data were fitted based on the Randles equivalent circuit (Figure 1A, inset), where R_s_ stands for solution resistance, C_dl_ is the double layer capacitance, and Z_w_ and R_ct_ represent the Warburg resistance and charge transfer resistance, respectively. As can be seen in Figure 1A, each electrode displays semicircles of varying diameters in the high frequency region, corresponding to R_ct_. The reaction occurring in the lower frequency linear region corresponds to the diffusion limited process. The R_ct_ value of the CPE is 491 Ω. After modification with 3DGS, the R_ct_ value decreased considerably to 278 Ω. This decrease in resistance is attributed to the conductive character of goethite as well as the enhanced surface area of 3DGS, offering higher electron conduction paths and accelerated electron transfer. The estimation of the heterogeneous electron transfer rate constant (k_s_) [58] is also of paramount interest for examining the electrode’s performance. This rate constant indicates the electron transfer speed between the analyte’s redox-active species and the solid electrode surface. The k_s_ is determined using the following equation, which incorporates several parameters including molar gas constant (R = 8.314 J·mol^−1^·K^−1^), absolute temperature (T = 298 K), number of electrons (n = 1), Faraday constant (F = 96,485 C·mol^−1^), geometrical area of the electrode (A = 0.125 cm^2^), and concentration of the potassium ferro- and ferricyanide solution (C = 5 mM):$$k_{S}=\frac{RT}{n^{2}F^{2}ACR_{ct}}$$

According to the equation, the values of k_s_ were estimated as 0.86 and 1.5 cm·s^−1^ for CPE and 3DGS, respectively, indicating a significant improvement in k_s_, which increased by approximately 1.74 fold after the electrode modification. This underscores the high efficacy of 3DGS in enhancing sensors’ electrochemical performance.

For the further electrochemical characterization of the modified electrode, CVs of CPE and 3DGS/CPE were recorded in 0.1 M KCl containing 5 mM [Fe(CN)_6_]^3−/4^ at a scan rate of 0.1 V·s^−1^ (Figure 1B). The ferro- and ferricyanide redox reactions on the CPE and 3DGS/CPE electrode surface displayed a quasi-reversible behavior with peak-to-peak separation of 250 and 200 mV, respectively, which are much higher than the ideal value of 57 mV, predicted for reversible single electron transfer reactions. However, the higher peak current value and the smaller peak-to-peak distance of hexacyanoferrate II/III anions at 3DGS/CPE compared to the CPE are attributed to the increase in the electrochemically accessible surface area (EASA) of the electrode and the facilitation of electron transfer [59].

To determine the EASA of the electrodes, CVs were obtained for both CPE and 3DGS/CPE at different scan rates (Appendix A) in 0.1 M KCl containing 5 mM [Fe(CN)_6_]^3−/4^. The EASA was then calculated using the Randles–Sevcik equation [60]:Ipa=2.69×105n32ACD12ν12
where I_pa_ is the anodic peak current, n is number of electrons, C is the potassium ferro- and ferricyanide solution concentration, D is the diffusion coefficient of the ferrocyanide ion and is expressed as 7.26 × 10^−6^ cm^2^·s^−1^ [61], and ν is the scan rate. The EASA was determined to be 0.107 cm^2^ for the CPE and 0.220 cm^2^ for 3DGS/CPE, obtained from the slope of the equation of I_pa_ as a function of ν^1/2^ (Appendix A). The increase in the EASA after electrode modification implies an improvement in the availability of active sites for redox reactions.

### 3.2. Electrochemical Response of Sensors towards DA and TRP Oxidation

To evaluate the sensing performance of the proposed sensor, CVs of bare CPE and 3DGS/CPE were recorded in 0.1 M phosphate buffer pH 6 in the absence and presence of 80 μM DA and TRP at a scan rate of 0.1 V·s^−1^ (Figure 2). As shown, 3DGS/CPE facilitates the electrochemical quasi-reversible oxidation of DA to dopamine o-quinone through a two-electron/two-proton process and the subsequent reduction of dopamine o-quinone to DA [23]. The I_pa_ of DA recorded at CPE and 3DGS/CPE increased from 2.6 μA to 5.3 μA and an anodic peak potential (E_pa_) shift occurred from 0.33 V to 0.32 V. Following the same trend, for the well-defined irreversible TRP oxidation to 2-amino-3-(5-oxo-3,5-dihydro-2H-indol-3-yl) propionic acid [62], the I_pa_ rose from 4.62 μA at CPE to 8.75 μA at 3DGS/CPE accompanied by a slight negative shift in the E_pa_ from 0.87 V to 0.84 V. Electrode modification with 3DGS significantly improves the I_pa_ and decreases the E_pa_ value, which means that oxidation of DA and TRP occurs more readily over the 3DGS/CPE. This confirms the electrocatalytic activity of 3DGS towards DA and TRP oxidation, as well as producing a higher heterogeneous electron transfer rate and enhanced adsorption of organic molecules on the surface of the electrode. Accordingly, the electrochemical features of the 3DGS/CPE are much desirable for the voltametric detection of DA and TRP. Hence, 3DGS/CPE is used as a working electrode for further investigations.

A prevalent challenge associated with electrochemical DA sensors is the formation of an insulating polydopamine film on the electrode surface during the application of positive potentials. It is crucial to note that the electropolymerization of DA on the electrode surface occurs under specific conditions, such as low scan rates (typically around 20 mV·s^−1^), extensive cycling (more than 15 cycles) and higher pH levels [63,64]. To ensure that DA electropolymerization exerts a negligible impact on sensor performance, CVs of 3DGS/CPE were recorded in a 0.1 M phosphate buffer at pH 6, containing 300 μM DA and TRP over 5 cycles (Appendix A). Under the optimized condition (pH 6, scan rate of 0.1 V·s^−1^), the I_pa_ for DA and TRP retained 80% of their initial values after 5 cycles.

### 3.3. Influence of pH on DA and TRP Oxidation

The impact of electrolyte pH on I_pa_ and E_pa_ of 30 μM DA and TRP at 3DGS/CPE in the pH range of 3–8 using 0.1 M phosphate buffer was investigated by DPV, and the results are shown in Figure 3A. The electrochemical oxidation of DA and TRP occurs through a deprotonation process induced by electron release, which is favorable at higher pH. Hence, the oxidation peak linearly shifts to lower E_pa_ values with increasing pH (Figure 3B). The I_pa_ of DA reaches the maximum value at pH 6 and gradually decreases as the pH increases (Figure 3C). Based on a study by Li et al. [65] the DA (p*K*_a_ 8.9) undergoes oxidation to dopamine *ortho*-quinone, which is relatively stable in an acidic environment. Under acidic conditions, the deprotonation of the amino group is inhibited, which hinders the intramolecular cyclisation process of the product [66]. The slope value of −76 mV from the linear regression equation for DA oxidation indicated that the electron transfer was accompanied by an equal protons’ transfer during the oxidation of DA.

TRP mainly exists as a cation in the pH range of 3–5 and as a zwitterion in the pH range of 5–8. Therefore, at lower pH, an attractive electrostatic force exists between the negatively charged 3DGS/CPE surface and the protonated amino group of TRP, which dominates over the π–π interaction [23]. The I_pa_ gradually decreases with increasing pH, and the highest I_pa_ value for TRP is obtained at pH 3. An increase in the I_pa_ at p*K*_a_ (2.38) and a significant decrease in the I_pa_ at p*K*_a_ (9.39) is expected for TRP [67]. As shown in Figure 3B, E_pa_ and pH are linearly dependent. The −38.8 mV slope suggested the involvement of an equal number of electrons and protons in this oxidation process. The ratio of the number of electrons and protons involved in the oxidation process of DA and TRP can be determined using the following equation [23]:ⅆEpⅆpH=−2.303mRTnF
where n and m are the number of electrons and protons. The obtained ratios are 1.2 and 0.7 for DA and TRP, respectively. To achieve a high current response for DA and considering the known phenomenon of neurotransmitter polymerization on the electrode surface at higher pH levels, as well as to maintain the pH of the electrolyte solution close to physiological conditions, a pH of 6.0 was selected [68].

### 3.4. Scan Rate Influence on the Oxidation of DA and TRP

The reaction kinetics of DA and TRP oxidation on the surface of 3DGS/CPE were further evaluated by CV measurements at different scan rates (20–350 mV·s^−1^) in 0.1 M phosphate buffer pH 6 containing 100 μM DA and TRP. As can be observed in Figure 4A, there is an increase in I_pa_ and a slight positive shift in the E_pa_ with an increasing scan rate. Figure 4B illustrates the linearity between the I_pa_ and the square root of the scan rate in the oxidation reaction of DA and TRP on 3DGS/CPE. The results indicate that the oxidation of DA and TRP is controlled by solute diffusion at the 3DGS/CPE interface [69]. In order to verify the diffusion controlled electrochemical process, the logarithm of anodic peak current (log I_pa_) vs. the logarithm of the scan rate (log υ) was plotted to obtain the theoretical slope value of 0.5 (Figure 4C). The obtained slope was 0.37 and 0.47 for DA and TRP, respectively. This observation revealed that the electrochemical oxidation of DA and TRP on the 3DGS/CPE surface is a diffusion-controlled electrode process [26].

The relationship between E_pa_ and the logarithm of the scan rate for DA and TRP is shown in Figure 4D. Based on the linear regression equations and the Laviron theory, the number of electrons involved in the electrochemical reaction was calculated to be 2.03 and 2.32 for DA and TRP, respectively. The proposed mechanism for electrochemical DA and TRP oxidation reaction at near-natural pH is shown in Figure 1.

The electron transfer coefficient (α) is also calculated from the slope values using the following equation:$$E_{pa}=\alpha +\frac{2.303RT}{\left(1-\alpha \right)nF}\log\nu$$

The obtained α value for the electrochemical oxidation of DA is 0.56, and 0.75 for TRP, which is well supported by the reported values [23,26,69]. These findings suggest that 3DGS/CPE has acceptable electrocatalytic activity for the detection of DA and TRP.

### 3.5. Individual Determination of DA and TRP

The analytical performance of 3DGS/CPE for the individual quantification of DA and TRP is evaluated using the DPV method. Figure 5A,C show the DPV curves of 0.1 M phosphate buffer pH 6 containing a mixture of DA and TRP, keeping the concentration of one species constant while changing the other analyte. As shown in Figure 5B, a linear dynamic range from 4 to 230 μM concentrations of DA in the presence of 20 μM TRP was obtained. The detection limit (LOD) for DA oxidation at 3DGS/CPE was calculated as low as 1.82 μM from 3 σ/m (σ is the standard deviation of the signal in blank solution, and m is the slope of the calibration curve for lower analyte concentrations) with a sensitivity of 0.9432 μA·cm^−2^·μM^−1^.

For TRP, the calibration curve in the range of 2–150 μM concentrations of TRP in the presence of 20 μM DA showed a linear response (Figure 5D). The LOD and the sensitivity were 0.85 μM and 2.016 μA·cm^−2^·μM^−1^, respectively. Based on these observations, accurate measurement of the individual DA and TRP in a mixture analyte without any significant deviation from the current is possible. In both cases, the interferent analyte’s current was almost constant, with less than 5% change.

### 3.6. Simultaneous Determination of DA and TRP

Simultaneous quantification of DA and TRP was performed in 0.1 M phosphate buffer pH 6 at 3DGS/CPE using DPV (Figure 6A). As can be seen, while the peak current increases by increasing the DA and TRP concentration, shift of the peak potentials towards more positive potentials is observed due to the generation of more electro-inactive oxidation species that block the electrode surface. The linear calibration curves for the peak current versus the concentrations of DA and TRP were obtained at 3DGS/CPE. Figure 6B illustrates the calibration curves for DA and TRP, which were observed between 4 and 246 μM for DA concentrations with a linear regression equation of I(μM) = 0.1188 C + 1.1595 (R^2^ = 0.9924), and for TRP concentrations between 2 and 150 μM with a linear regression equation of I(μM) = 0.2331 C + 1.8659 (R^2^ = 0.9905). The sensitivity of the prepared DA and TRP sensor based on 3DGS/CPE was calculated to be 0.9504 and 1.8648 μA·μM^−1^·cm^−2^, respectively. The LOD values of 3DGS/CPE were obtained as low as 1.9 for DA and 0.37 μM for TRP. It should be noted that the obtained slopes in this method are very close to those obtained in individual measurements, which indicates that the measurement of each compound is free from interference from the other. Upon comparing the data presented in Appendix A [70,71,72,73,74,75,76,77,78], it is evident that the 3DGS CPE offers enhanced or equivalent performance in the simultaneous determination of DA and TRP.

### 3.7. Investigation of Stability, Repeatability, Reproducibility, and Selectivity of 3DGS/CPE

The precision and reproducibility of 3DGS/CPE was evaluated to assess the analytical efficiency and applicability of the proposed method. The fabricated electrode was stable in a solution of 0.1 M KCl containing 1 mM [Fe(CN)_6_]^3−/4−^ over 100 cycles of CV, and the I_pa_ and I_pc_ remained at 85% of their initial values (Appendix A). To test the reproducibility, the DPV response of four fresh electrodes in a solution of 0.1 M phosphate buffer pH 6 containing 150 μM DA and TRP was recorded (Appendix A). The relative standard deviation (RSD) of the electrochemical response of the DA and TRP was calculated to be 4.62 and 3.34%, respectively. In order to investigate the repeatability of 3DGS/CPE, four successive measurements of 30 μM DA and TRP using the same electrode showed an RSD of 7.09 and 2.81%, respectively (Appendix A). These observations indicated that the fabricated sensor has an excellent reproducibility and repeatability for continuous analysis.

The long-term stability of the sensor was evaluated by DPV measurements of the 3DGS/CPE in the presence of 30 μM DA and TRP. The electrode was kept at room temperature and supported with a lid. The I_pa_ of 30 μM DA and TRP were measured over one month (Figure 7A). After one month, the I_pa_ was still maintained at 78% and 85% of its original value for DA and TRP, respectively.

The selectivity of the sensors in the biological samples is very crucial for the real quantification of the analytes. Hence, the DPV of 0.1 M phosphate buffer pH 6 containing 30 μM DA and TRP was performed in the presence of other commonly interfering molecules. The I_pa_ and E_pa_ of DA and TRP were observed to not be significantly interfered with in the presence of 100 fold of potentially interfering substances (NaCl, KCl, glucose, alanine, and CaCl_2_) (Figure 7B). The same amount of AA, UA and ML caused a slight positive potential shift and did negligibly interfered with DA and TRP (signal change in DA and TRP is less than 5%) (Figure 7C). Therefore, these organic compounds have a minimal effect on the accuracy of simultaneous DA and TRP measurement.

### 3.8. Determination of DA and TRP in Real Samples

The fabricated 3DGS/CPE was utilized for simultaneous quantification of DA and TRP in human urine samples using the standard addition method. The real sample was directly added to 0.1 M phosphate buffer pH 6 without any pre-treatment and then spiked with standard DA and TRP solutions, followed by recording their corresponding differential pulse voltammograms at a scan rate of 0.1 V·s^−1^ (Figure 8). The results are shown in Table 1. Based on the recovery calculations, it has been observed that the fabricated 3DGS/CPE achieves satisfactory recovery rates ranging from 101 to 116.44%, with an RSD below 5%. Therefore, the suggested sensor can be suggested for the simultaneous DA and TRP quantification in human urine sample.

## 4. Conclusions

Herein, a 3D goethite–spongin-modified carbon paste electrode (3DGS/CPE) is fabricated for the individual and simultaneous quantification of the biological molecules, including DA and TRP. The electrochemical results indicated that the 3DGS/CPE could effectively improve the electron transfer kinetics and exhibited high electrocatalytic activity towards DA and TRP. An important highlight of this method is the first-time report of the 3D goethite–spongin integrated into an electrode structure for simultaneous determination of DA and TRP. The robustness of the analytical procedure, the acceptable sensitivity, and the wide linear concentration range of the DA and TRP on the 3DGS/CPE are highly desirable. The fabricated sensor showed an excellent selectivity towards DA and TRP in the presence of various interfering molecules. The proposed sensor is analytically applicable for the quantification of these molecules in human urine samples.

## Data Availability

The original contributions presented in this study are included in this article and Appendix A, further inquiries can be directed to the corresponding authors.

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
