# Peer review of "Simultaneous Electrochemical Detection of Dopamine and Tryptophan Using 3D Goethite–Spongin Composites"

_biomimetics, 2024, doi:10.3390/biomimetics9060357_

Round 1
Reviewer 1 Report
Comments and Suggestions for Authors
Author Response
This manuscript reported a 3D goethite-spongin composite based electrochemical sensor, simultaneously detecting dopamine and tryptophan. The manuscript shows some major issues need to be addressed before it can be published in biomimetics. Specially, the comments are as following:
Line 172: The decrease in Rct value may be attributed to the presence of graphite powder. It is advisable to conduct the electrochemical impedance spectroscopy (EIS) data for the graphite powder/carbon paste electrode (CPE) system to substantiate the assertion that "T This decrease in resistance is attributed to the conductive character of goethite as well as the enhanced surface area of 3DGS, offering higher electron conduction paths and accelerated electron transfer. "
We would like to sincerely thank you for the pre-review of this manuscript. As has been explained in section 2.3., bare CPEs inherently are made from graphite powder and paraffin oil. Therefore, the EIS method has already been employed in our manuscript to investigate the behaviour of the 3DGS composite.
Line 183, please check the equation; it pertains to the standard rate of electron transfer (k0) as cited in your manuscript. (https://doi.org/10.1016/j.electacta.2018.08.021)
We agreed. It was edited.
Line 197, the calculation of Effective Electrochemical Surface Area (EASA) appears erroneous. It is noted that the Randles–Sevcik equation, formulated for reversible surface reactions, is not applicable to the irreversible electrode system elucidated earlier.
In other to calculate the EASA of the fabricated electrode, the CVs recorded in 0.1 M KCl containing 5 mM [Fe(CN)6]3-/4-. As illustrated in Figure S1, the reaction is diffusion-controlled and exhibits good reversibility on the 3D goethite-spongin modified carbon paste electrode (3DGS/CPE). Consequently, in line with reference number 60, it is affirmed that the peak current is proportional to the electroactive surface area of the working electrode and can be appropriately expressed by the Randles-Sevcik equation.
In Figure 3 A, it is recommended to replenish the pH values corresponding to the curves.
We appreciate your comment. However, the Authors have opted to retain the original presentation of the figure.
Figure 4 C suggests reversibility within the system, while Figure 4 D uses the Laviron theory, primarily applicable to irreversible systems. This presents a contradiction in the interpretation.
The apparent contradiction between Figure 4C suggesting reversibility and the use of the Laviron theory in Figure 4D can be reconciled by considering the behaviour of the system at higher scan rates. Based on this reference (doi: 10.1016/j.ccr.2009.12.023), at higher scan rates, the Epa vs. log n should be a straight line where the slope of the line in proportional to α. Additionally, it is well-documented that at higher scan rates, electrochemical systems tend to exhibit irreversible behaviour. Therefore, the application of the Laviron theory in Figure 4D is justified in analysing the system's behaviour under these conditions.
Reviewer 2 Report
Comments and Suggestions for Authors
The manuscript topic belongs to intensively studied, and dozens of other electrocatalytical systems were suggested. Therefore:
(i) the authors should make analysis of literature and indicate clearyl the novelty;
(ii) the sensor performance should be compared mit dozens of other publication on this topic;
(iii) the usual problem of electrochemical dopamine sensors is a formation of insulating film of polydopamine on the electrode. Is it the case for the current material?
(iv) the usual selectivity problem of electrochemical dopamine sensors is the interference with ascorbic acid and oric acid. Has it been tested?
Author Response
The manuscript topic belongs to intensively studied, and dozens of other electrocatalytical systems were suggested. Therefore:
(i) the authors should make analysis of literature and indicate clearly the novelty;
We would like to thank you for the thoughtful comments and efforts towards improving our manuscript. It is worth noting that the topic of our manuscript is within a field that has been extensively researched. In our study, we have demonstrated the simultaneous detection of DA) and TRP using a novel biomimetic material with a 3D structure without using biomolecules as recognition elements. Notably, to date, there have been no reports of similar studies in the literature.
(ii) the sensor performance should be compared with dozens of other publication on this topic;
We have included Table S1 in the supporting information file, which provides a comparison of the analytical parameters of our study with those of other reported electrochemical sensors for the simultaneous detection of DA and TRP.
(iii) the usual problem of electrochemical dopamine sensors is a formation of insulating film of polydopamine on the electrode. Is it the case for the current material?
Our experimental observations indicate that under our optimized conditions, the electropolymerization rate of polydopamine is notably reduced. This can be attributed to the absence of an inert gas atmosphere and the slightly acidic pH of 6 utilized in our experiments. Furthermore, the application of higher scan rates further contributes to minimizing polydopamine electropolymerization. Hence, we can assert that the formation of an insulating polydopamine film on the electrode surface is not a prominent concern for the current study.
(iv) the usual selectivity problem of electrochemical dopamine sensors is the interference with ascorbic acid and uric acid. Has it been tested?
The sensor's selectivity has been assessed in the presence of both ascorbic acid and uric acid. The findings of these tests are depicted in Figure 7C.
Round 2
Reviewer 1 Report
Comments and Suggestions for Authors
The manuscript can be accepted in present form.
Author Response
Thank you for your time.
Reviewer 2 Report
Comments and Suggestions for Authors
Unfortunately, the authors did not take into account any my comment and suggestion.
The manuscript cannot be accepted (not only because of scientific but also because of ethical reasons) if the authors ignore numerous scientific literature in the field.
The inserted Table 1 presents only a very-very small part of the literature.
The answer concerning dopamine polarization is not convincible. It is not clear, why the absence of inert gas atmosphere can help. The authors write on the high scan rate, but this statement is not true. Actually, the authors should show many cycles of CV in the presense of dopamine - has it been done?
Author Response
We deeply regret the misunderstanding and have tried to address the comments more clearly and revised the manuscript accordingly. We hope these revisions address your concerns and enhance the scientific rigor and ethical standards of our work. Thank you for your valuable feedback.
-We faced a challenge when comparing this manuscript with other studies for the simultaneous detection of dopamine (DA) and tryptophan (TRP) due to the variety of methods and interfering agents commonly used in such studies. In this study, we first evaluated the analytical performance of 3DGS/CPE for the individual quantification of DA and TRP in a mixture of analytes in the presence of an interfering analyte (section 3.5). Subsequently, the simultaneous quantification of DA and TRP was performed by increasing the concentrations of DA and TRP (section 3.6).
Given the concept of our study, it is not reasonable to compare our work with studies that solely measure TRP or DA individually, as our study includes the individual determination of these compounds in the presence of an interferent in a mixture. The interferent’s concentration even effect the linear range and LOD of sensors.
Additionally, many similar studies have not conducted individual detection experiments in a mixture.
When comparing sensor performance for the simultaneous detection of DA and TRP, it is true that there are many studies available, but most of them also detect another compound such as uric acid, ascorbic acid, acetaminophen or serotonin along with DA and TRP simultaneously. The performance of the sensors is influenced by the number of markers and the concentration of interfering compounds in the mixture.
Therefore, comparing these types of studies can be quite challenging due to the differing experimental conditions.
Nonetheless, the novelty of our study lies in the use of a biopolymer as a substrate for the formation of goethite, which has a catalytic activity and offers an advantage over metal nanoparticles due to its simple synthesis and environmental friendliness.
We have added more studies that simultaneously measure DA and TRP to Table S1. We have focused on comparing our work with studies that aim for the simultaneous detection of these compounds while also being capable of their individual detection in a mixture.
-Regarding the polymerization of dopamine, our statement about the effect of an inert gas is referenced from:
[- https://doi.org/10.1016/j.eurpolymj.2022.111346] which again is quoted here:
(‘’ Coating with Polydopamine was applied for the working electrodes … Moreover, an inert gas (argon or nitrogen) atmosphere is crucial, as the oxygen is forming a radical anion (superoxide, O2−) within a reversible one-electron reduction process.’’)
Even though the polymerization of neurotransmitters on the electrode surface is a well-known phenomenon at higher pH, we used a phosphate buffer saline solution at pH 6. [https://doi.org/10.1021/acs.analchem.9b02967].
Additionally, it should be noted that the previous statement regarding the effect of scan rate on DA electropolymerization was not accurately adjusted. Based on some studies, it can be concluded that at lower concentrations of DA, electropolymerization occurs at lower scan rates [https://doi.org/10.1002/masy.201400130, http://dx.doi.org/doi:10.1016/j.snb.2016.01.012]. However, there are also numerous reports indicating that DA electropolymerization occurs at higher scan rates, which cannot be disregarded.
Moreover, it is evident that electro-polymerization is inevitable in the presence of a catalyst. However, as previously mentioned, we can mitigate this effect based on our experimental conditions. It is important to note that this issue is not extensively investigated in the majority of studies, as it does not negatively impact the sensor's performance. In fact, many studies utilize polydopamine-modified electrodes for dopamine detection, demonstrating their efficacy. Relevant references include:
[- https://doi.org/10.1016/j.jelechem.2021.115133
- https://doi.org/10.1039/C7AY00991G
- https://doi.org/10.1016/j.msec.2019.110602]
We have recorded cyclic voltammetry for five cycles in the presence of 0.5 mM of DA and TRP which is relatively high concentration (Figure 1). The CV results after five cycles indicate that the electrode surface maintains satisfactory performance. For comparison, Figure 2, extracted from the following reference, shows that insulation of the electrode surface occurs at higher cycle numbers [https://doi.org/10.1002/masy.201400130] . Figures are attached to this file.
Based on this evidence, we assert that the sensor's performance is not significantly affected by DA electro-polymerization, which is an inevitable phenomenon in DA sensors. This polymerization does not adversely impact the sensor's overall performance.

Round 3
Reviewer 2 Report
Comments and Suggestions for Authors
-